# Effects of Non-Surgical Periodontal Therapy on Glycemic Control in Prediabetes and Diabetes Patients with Stage II–IV Periodontitis as Monitored by Active-Matrix Metalloproteinase-8 Levels

**DOI:** 10.3390/biomedicines13040969

**Published:** 2025-04-16

**Authors:** Kehinde Adesola Umeizudike, Solomon Olusegun Nwhator, Olayiwola Ibrahim Olaoye, Ayodele Charles Ogundana, Ismo T. Räisänen, Olufemi Adetola Fasanmade, Oladunni Ogundana, Obiefuna Ajie, Timo Sorsa

**Affiliations:** 1Department of Preventive Dentistry, Faculty of Dental Sciences, College of Medicine, University of Lagos, Idi-araba, Lagos P.M.B. 12003, Nigeria; ogundana.ayodele@gmail.com; 2Department of Oral and Maxillofacial Diseases, Head and Neck Center, University of Helsinki and Helsinki University Hospital, 00290 Helsinki, Finland; ismo.raisanen@helsinki.fi (I.T.R.); timo.sorsa@helsinki.fi (T.S.); 3Department of Preventive and Community Dentistry, Faculty of Dentistry, College of Health Sciences, Obafemi Awolowo University, Ile-Ife 220005, Nigeria; sonwhator@oauife.edu.ng; 4Department of Preventive Dentistry, Lagos State University Teaching Hospital, Ikeja, Lagos 100271, Nigeria; olayiwola2olaoye@gmail.com; 5Department of Medicine, Faculty of Clinical Sciences, College of Medicine, University of Lagos, Idi-araba, Lagos P.M.B. 12003, Nigeria; ofasanmade@unilag.edu.ng; 6Department of Oral and Maxillofacial Pathology and Oral Biology, Faculty of Dental Sciences, College of Medicine, University of Lagos, Idi-araba, Lagos P.M.B. 12003, Nigeria; dunnidana23@yahoo.com; 7Department of Clinical Pathology, Faculty of Clinical Sciences, College of Medicine, University of Lagos, Idi-araba, Lagos P.M.B. 12003, Nigeria; oajie@unilag.edu.ng; 8Division of Oral Diseases, Department of Dental Medicine, Karolinska Institutet, 17177 Stockholm, Sweden

**Keywords:** active-matrix metalloproteinase-8, type 2 diabetes mellitus, prediabetes, glycated hemoglobin, periodontitis

## Abstract

**Background/Objectives**: Previous research indicates that non-surgical periodontal therapy (NSPT) improves glycemic control in individuals with prediabetes and diabetes who have periodontitis. Few studies have demonstrated its effects on mouthrinse active-matrix metalloproteinase-8 (aMMP-8) levels as it relates to glycemic control. We assessed the periodontal treatment response of stage II–IV periodontitis patients with prediabetes, diabetes, and normoglycemia, regarding glycated hemoglobin (HbA1c) and mouthrinse aMMP-8 levels using point-of-care kits (PoC). **Materials and Methods**: Eighty-eight adults (11 normoglycemic, 32 prediabetic, 45 with type 2 diabetes), aged 25–78, with stage II–IV periodontitis were included. Full-mouth clinical examinations were used to evaluate their periodontal parameters. HbA1c and mouthrinse aMMP-8 levels were assessed using PoC kits before and approximately three months after scaling and root planing. **Results:** There were positive treatment effects of non-surgical periodontal therapy on periodontal clinical parameters, aMMP-8 and HbA1c levels in the prediabetes and diabetes groups. The aMMP-8 reduction was significant (*p* < 0.001) in the prediabetes and prediabetes + diabetes groups, while HbA1c decreased significantly in the diabetes and prediabetes + diabetes (*p* < 0.001) groups. In contrast, a non-significant increase in mean aMMP-8 levels, HbA1c, and CAL was observed in normoglycemia (*p* > 0.05). Stage III + IV periodontitis showed significant treatment effects for aMMP-8 (*p* < 0.001) and HbA1c (*p* < 0.01) compared to stage II, regardless of glycemic status. **Conclusions:** Non-surgical periodontal therapy significantly improves periodontal health as well as HbA1c and aMMP-8 levels in people living with prediabetes and diabetes.

## 1. Introduction

Periodontitis and diabetes are chronic, highly prevalent, interconnected, non-communicable inflammatory diseases that present significant public health challenges globally [1,2,3]. According to the International Diabetes Federation (IDF), diabetes mellitus (DM) affects 537 million individuals worldwide, with three-quarters of adults living in low- and middle-income countries [4]. In Nigeria, approximately 11.2 million adults (5.8%) have diabetes [5], and 15.8 million (13.2%) have prediabetes (impaired fasting glucose) [6]. People living with diabetes and prediabetes have comorbidities that overburden healthcare facilities [7]. The bi-directional relationship between the development of diabetes, its pathogenic pro-inflammatory processes, and periodontitis, remains the focus of extensive research. Severe periodontitis affects 23.6% of adults globally [8], with a notably higher prevalence among diabetics (49.1%), compared to non-diabetics (31.8%) [9].

Consequently, DM increases the prevalence, extent, and severity of periodontitis, with the tissue destruction resulting from the interaction between the oral biofilm dysbiosis and the host’s immune mechanisms [3]. Type-2 diabetes mellitus (T2DM) leads to an elevated accumulation of advanced glycation end products (AGEs) in the periodontal tissues due to the non-catalytic glycation of molecules such as proteins, lipids, and nucleic acids [10]. The interaction between AGEs and their receptors (RAGE), primarily found on macrophages, facilitates the activation of the local immune and inflammatory responses by an adipose pathway [10,11]. These upregulated responses increase the secretion of pro-inflammatory cytokines, including interleukin (IL)-1β, tumor necrosis factor-α (TNF-α), and IL-6, which cause oxidative stress and a disruption of the receptor activator of NF-κB ligand/osteoprotegerin (RANKL/OPG) axis to favor bone resorption [12,13,14,15].

In addition to cytokines, proteolytic enzymes, particularly matrix metalloproteinases (MMPs), are critical proteases that contribute to the periodontal tissue degradation [16,17,18]. MMP-8, referred to as collagenase-2 or neutrophil collagenase, primarily secreted by neutrophils, serves as the major host-derived collagenase, capable of degrading type I collagen in the periodontium and cleaving tri-helical collagen [19,20,21]. However, non-PMN-lineage cells such as resident fibroblasts and endothelial cells can also express it in lesser quantities. Physiologically, MMP-8 mainly facilitates neutrophil migration from the circulation to the gingival sulcus via a chemotactic gradient. Pathologically, this function eventually damages soft and hard tissues in periodontitis [20,22]. In fact, the activated form of matrix metalloproteinase-8 (aMMP-8) currently represents one of the most promising oral fluid biomarkers for periodontitis in both its early and advanced stages [23]. Active MMP-8 is strongly correlated with an increased severity of periodontal disease, from gingivitis > P-stage III > P-stage IV [24], and a higher diagnostic accuracy for periodontal disease than health [24,25,26,27,28]. Therefore, the aMMP-8 PoC test can be conveniently utilized to monitor the progression of periodontitis, with a threshold of 20 ng/mL [25], and diagnose periodontitis in prediabetic patients [29]. The expression of MMP-8 in gingival tissues and oral fluids can also be upregulated by the development of diabetes from metabolic syndrome via prediabetes [30]. This upregulation of MMP-8 can thus accelerate periodontal tissue destruction [31,32]. The impact of diabetes on periodontal disease is demonstrated by the significant correlation between periodontitis and salivary MMP-8 activity [33].

MMP-8 also accelerates the progression of diabetes by cleaving and deactivating the insulin receptor. However, synthetic inhibitors of MMP-8, such as doxycycline, can help to mitigate this effect [34]. Consequently, MMP-8 downregulation may prove beneficial in future diabetic treatment [34]. Non-surgical periodontal therapy (NSPT) has also been shown to be very effective in reducing aMMP-8 levels [35].

As periodontitis and diabetes are so clearly linked bilaterally [36,37], several studies have explored the clinical benefit of periodontal therapy on glycemic control in diabetes and prediabetes [38,39]. In addition, the application of antibacterial agents, including chlorhexidine, which is also an aMMP-8 inhibitor [40], can reduce the collagenolytic tissue destruction and facilitate healing [41]. This therapy can be supported by other adjuncts such as low-dose doxycycline, antibiotics, ozone therapy, laser therapy, dual-light photodynamic therapy, and fermented lingonberry mouthwash [42,43,44,45,46]. In addition, Pardo et al. [47] recently reported the application of topical biofilm disaggregating agents during NSPT to improve clinical and microbiological outcomes.

Given the burden of diabetes and its multisystemic complications, any therapy that can improve glycemic control by reducing glycated hemoglobin (HbA1c) is of benefit. There are limited studies reporting the effect of NSPT on both aMMP-8 and HbA1c in real-time, using chairside kits, among Nigerian patients living with prediabetes and diabetes. Our hypothesis was that NSPT will reduce aMMP-8 and HbA1c levels in diabetic and prediabetic patients with periodontitis compared to normoglycemic patients. This study aimed to assess the effect of NSPT on PoC aMMP-8 and HbA1c levels, as well as periodontal clinical parameters in periodontitis patients living with prediabetes and diabetes.

## 2. Materials and Methods

### 2.1. Study Design and Setting

The current study was a short-term longitudinal cohort study conducted at the Periodontology and the Endocrinology Clinics of the Departments of Preventive Dentistry and Medicine, respectively, at the Lagos University Teaching Hospital (LUTH), Idi-Araba, Lagos, South-West zone of Nigeria. The study was conducted from January 2022 to October 2023.

### 2.2. Study Groups and Clinical Assessment

The sociodemographic characteristics of the participants in Table 1, including the age, gender, education, ethnicity, and religion, were recorded to give an overview of the multi-cultural background of the study participants and to assess the distribution of the study findings.

The highest level of education attained was categorized into primary, secondary, and tertiary; ethnicity was based on the three major ethnic groups in Nigeria: Yoruba, Igbo, and Hausa; and religion was grouped into Christianity and Islam. The inclusion criteria were adults ≥ 18 years of age, diagnosis of periodontitis, and the presence of at least 20 standing natural teeth. Periodontitis was defined and staged according to the new 2017 classification system as at least two or more interproximal sites with a clinical attachment level (CAL) of ≥2 mm on two non-adjacent teeth [48]. Participants were thus staged into II, III, and IV after enrollment (Figure 1). Current smokers, pregnant women, patients on antibiotic therapy or prior periodontal treatment in the preceding three months to the study, and patients on immunosuppressants or non-steroidal anti-inflammatory drugs were excluded from the study. All participants were tested using a hemoglobin A1C (HbA1c) kit and an aMMP-8 point of-care (PoC) test kit with a cut-off of 20 ng/mL [25,49]. Full-mouth clinical examinations were performed to assess their periodontal and oral health status. The diabetic patients all had T2DM. Most of the T2DM and prediabetes patients were consecutively enrolled from the endocrinology clinic and the normoglycemic patients from the periodontology clinic.

### 2.3. Sample Size Calculation

A sample size calculation for a paired *t*-test to evaluate the treatment effect of NSPT was performed (G*Power 3.1), which revealed that a total of 27 patients were required to reach 80% power with a medium effect size 0.50 [50] and a significance level of 5%. This was added by 10% compensation for possible missing data, which resulted in a total sample size of 30. Based on Keskin et al. [50], we estimated/expected a medium-to-large effect size for this study and used a medium effect size to not underestimate the sample size.

### 2.4. Periodontal Clinical Parameters

The parameters evaluated in the current study included the periodontal probing depth (PPD), clinical attachment level (CAL), simplified oral hygiene index (OHI-S), and percentage bleeding on probing (BOP%). To rule out any bias, two examiners were calibrated to perform periodontal examinations using a Williams periodontal probe. The inter-examiner reliability yielded a kappa score of 0.76. The PPD was defined as the distance between the gingival margin and the base of the pocket. The distance was assessed by inserting the probe as much as possible parallel to the long axis of the tooth. The CAL was calculated as the distance between the cemento-enamel junction and the base of the pocket. The PPD and CAL were measured at six sites per tooth (mesiolingual, lingual, distolingual, mesiobuccal, buccal, and distobuccal). The CAL was computed by adding the PPD reading to the gingival margin level when recession was present and subtracting the gingival margin level from the PPD reading when the gingival margin was coronal to the cemento-enamel junction. Periapical radiographs and/or orthopantomographs were used by the examiners to confirm alveolar bone loss in the affected teeth.

### 2.5. aMMP-8 Point-of-Care (PoC) Mouthrinse Testing

The aMMP-8 PoC is a safe, rapid, non-invasive diagnostic test that detects a biomarker, aMMP-8 levels for periodontal diseases, using a simple mouthrinse sample. It has been validated and standardized through numerous researches and is commercially available as a lateral flow immunotest stick that is visually (+, −) and quantitatively read by the ORALyzer^®^ Digital Reader System (Dentognostics GmbH, Jena, Germany), with a cut-off of 20 ng/mL and a 30 s pre-rinsing [25,26]. In the current study, the mouthrinse sample was collected before the comprehensive periodontal examination, and participants were instructed not to eat, drink, or brush their teeth at least 1 h before sampling. They were asked to rinse with drinking water for 30 s and then spit the water out. After a 60 s interval, the patients were provided with 5 mL of distilled water from the aMMP-8 test kit to rinse again for 30 s, which was then expectorated into a measuring cup. Following the manufacturer’s instructions, 3 ml of the mouthrinse sample was drawn from the cup into a syringe and filtered, and 3 drops were placed in a well of the aMMP-8 Periosafe^®^ kit test cassette. The aMMP-8 level was quantitatively analyzed by the digital reader within five minutes, which displayed the results as in previous studies [32,49]. The test result was considered positive if aMMP-8 levels were ≥20 ng/mL. The mouthrinse aMMP-8 levels were assessed at baseline and at three months’ recall after NSPT.

### 2.6. Glycated Hemoglobin (HbA1c) Testing

All patients had their blood glucose screened to check their glycemic status by testing for HbA1c levels using the Anbio^®^ (Fluorescence Immunoassay) FIA HbA1c Rapid test kit (Anbio biotechnology, Frankfurt, Germany). The participants’ whole blood samples were collected through standard phlebotomy procedures in an EDTA sample collection bottle, a disposable pipette was then used to collect blood to 10 µL and added to the buffer tube and mixed thoroughly for 1 min, and 50 µL of the resulting sample mixture was then taken with a pipette and loaded onto the sample well of the test cartridge in the Anbio^®^ FIA test kit for quantitative analysis. The HbA1c was categorized into 3 groups based on the pre-treatment levels according to the International Diabetes Federation (IDF): normoglycemia (<5.7%), prediabetes (5.7–6.4%), and diabetes (≥6.5%) [4].

### 2.7. Non-Surgical Periodontal Therapy (NSPT)

Following an initial periodontal examination, the participants received NSPT in the form of full-mouth disinfection (scaling and root planing (SRP) of all sites with PPD ≥ 4 mm under local anesthesia, using chlorhexidine mouthrinse for irrigation), and oral hygiene instructions were given. The oral hygiene home care was to optimize the outcome of the NSPT and minimize biofilm buildup by ensuring patient compliance.

Participants with pre-HbA1c levels ≥ 6.5% (T2D) were further placed on adjunctive systemic antibiotics, namely amoxicillin 500 mg and metronidazole 400 mg eight hourly for five days because of their immunocompromised status and the severity of the periodontitis. The participants were recalled after 3 months to reassess the periodontal parameters (PPD, CAL, OHI-S, BOP%, mouthrinse aMMP-8, and HbA1c levels).

### 2.8. Statistical Methods

For analyzing the demographic data, continuous data were summarized as the mean (standard deviation), while discrete (categorical) data were presented as percentages. The continuous variables were compared by ANOVA, and categorical variables were compared by Fisher’s exact test, since it functions well no matter what sample size is used. The parameters analyzed for the study were aMMP-8, HbA1c (%), number of PPD sites, CAL, OHI-S, and BOP (%). The treatment effect (pre- and post-SRP treatment) on aMMP-8 (ng/mL), HbA1c (%), and periodontal parameters were evaluated by paired-samples *t*-test. Based on the central limit theorem that sample sizes of at least 30 can be assumed to be normal, parametric *t*-tests for paired/independent samples were used. Fisher’s exact test was used for categorical variables in this study, since it functions well no matter what sample size is used. The level of statistical significance was set at *p*-values ≤ 0.05. The quantitative data were analyzed using Statistical Product and Service Solutions (SPSS package, version 29.0.2.0).

## 3. Results

### 3.1. Sociodemographic Characteristics

The sociodemographic characteristics of the patient population is shown in Table 1. There were 46 males and 42 females, aged 25 to 78 years, with a mean age (SD) of 57.3 (13.2) years. However, the age of the diabetes group was higher than the normoglycemic and prediabetes groups (*p* < 0.05). Education, ethnicity, religion, and oral hygiene practice were not significantly associated with glycemic status.

### 3.2. Effect of Non-Surgical Periodontal Therapy (NSPT)

Periodontal parameters such as PPD ≥ 4 mm and OHI-S showed a significant reduction following NSPT for all patients, and CAL showed a non-significant reduction (Table 2). Paired-sample comparisons of mouthrinse aMMP-8 and HbA1c levels, before and after SRP, revealed that the mean aMMP-8 and HbA1c levels were reduced significantly (*p* < 0.001) (Table 3). There was a significant positive treatment effect in aMMP-8 and HbA1c levels after periodontal therapy in the entire total population, with a mean difference of 10.5 ng/mL and 0.4%, respectively. When adjusted for periodontitis staging, the reduction in aMMP-8 or HbA1c levels in stage II periodontitis did not reach the level of significance (*p* = 0.055 and *p* = 0.107, respectively) but attained statistical significance in stage III and IV (*p* = 0.004 and *p* = 0.002, respectively) (Table 3).

When adjusted for glycemic status, the normoglycemic group did not respond to the treatment as positively as patients with prediabetes and diabetes in aMMP-8 (ng/mL) and HbA1c (%) (Figure 2A,B), but there was a significant reduction in OHI-S levels (Figure 2C), PPD ≥ 4 mm in the prediabetes and diabetes groups and none in the normoglycemic group (Figure 3A). Only the diabetes group had a significant reduction in the CAL (Figure 3B), while the BOP (%) was reduced significantly in all glycemic groups (Figure 3C).

## 4. Discussion

The key findings of this study were that NSPT significantly improved glycemic control and was reflected by the significant reduction in HbA1c levels. Furthermore, aMMP-8 levels and periodontal clinical parameters were significantly reduced. However, each of the glycemic groups responded differently to the treatment. Generally, the treatment effect on HbA1c was statistically significant mainly in the diabetes group in pairwise comparison, as well as among stage III and IV periodontitis patients. There was a significant reduction in PPD ≥ 4 mm, OHI-S, and BOP (%) following periodontal treatment in all the glycemic groups but when adjusted for glycemic status, this was confined to the prediabetes and diabetes groups alone, with significant reduction in BOP (%) scores in all three glycemic groups.

The reduction in HbA1c levels in the current study following NSPT corroborate previous studies [3,51,52,53]. An absolute reduction of 0.43% in HbA1c levels, 3–4 months after the treatment of periodontitis in diabetes patients, was documented in a Cochrane systematic review [54]. This can be attributed to the effect of root surface debridement and optimized oral hygiene, which decreases both periodontal inflammation and the levels of circulating bacteria and bacterial products. In addition, reduced systemic levels of pro-inflammatory cytokines and mediators such as TNF-α and C-reactive protein lead to improvements in insulin resistance and insulin signaling [13]. This positive treatment effect in the diabetic group could also be attributed to the adjunctive use of the systemic antibiotics, as it helps to reduce the bacterial load in the periodontal pockets, particularly in deep or difficult-to-access periodontal sites. This is supported by studies that have combined SRP with systemic antibiotics and reported an improvement in periodontal parameters, including PPD and BOP in diabetic patients [55,56].

The current study found a mean 0.4% drop in HbA1c levels in all glycemic categories, and 87.5% of participants had prediabetes and diabetes, as in the Cochrane study. A recent randomized trial also observed a decrease in HbA1c levels following the application of natural extracts-based gel (home remedies) in type 1 DM patients, while the probiotics-based agents had no such influence [57].

In the current study, stage III and IV periodontitis had a substantially higher mean HbA1c reduction (0.42%) than stage II (0.3%). Moreover, the glycemic status significantly influenced the treatment response of HbA1c, evidenced by its decrease within the diabetes cohort. There was also a non-significant reduction of HbA1c in the prediabetes group. The benefit of periodontal therapy on glycemic control is more obvious in higher than lower baseline HbA1c levels [51,58]. Higher HbA1c levels have also been demonstrated in stage III and IV periodontitis patients with previously undiagnosed diabetes [59]. Research shows that NSPT lowers local inflammatory markers in periodontal pockets and systemic inflammation. This affects the insulin receptors, resulting in decreased hemoglobin glycosylation and HbA1c. Marconcini et al. [60] found a 0.52% drop in HbA1c three months following SRP but no further decrease at six months. This disparity in the impact of periodontal therapy on glycemic control may be due to the excessive inflammatory response experienced by patients with poor glycemic control due to angiopathy and impaired healing. Thus, periodontal therapy may reduce serum inflammatory markers and facilitate the remission of glucolipid metabolism and insulin resistance [51].

The non-significant treatment effect in normoglycemic patients is also supported by Nishioka et al. [61], who found a similar effect on markers of insulin and glucose metabolism among individuals with a lower baseline HbA1c of 5.6% compared to those with higher baseline HbA1c. Thus, periodontal therapy may exert optimal benefits in diabetic patients with higher baseline HbA1c who have poor glycemic control [51].

Systemic inflammation reduces following periodontal treatment in individuals with diabetes and periodontitis [62]. Thus, clinically meaningful HbA1c reductions can minimize diabetes complications such as micro- and macroangiopathy [63]. A linear link exists between HbA1c levels and diabetic microangiopathy risk, with a 1% reduction cutting the risk by 35% [51,64]. Periodontal therapy improves diabetes control better when the baseline HbA1c is higher [51]. The reduction in the HbA1c levels in the diabetic patients after the periodontal treatment is likely to have clinically meaningful effects on their systemic health. Clinicians should therefore take this into account and promptly refer patients with poor glycemic control to the dentist for comprehensive periodontal assessment and treatment [57].

The aMMP-8 levels were conveniently on-line and real-time visually, quantitatively, rapidly, and objectively assessed at the chairside using the PoC test kits. The notable decrease in aMMP-8 after NSPT in all the glycemic groups in our current study alludes to the already established role of aMMP-8 as a biomarker of periodontitis and its reduction following periodontal therapy [35]. However, when adjusted for glycemic status, the prediabetes and diabetes groups experienced significantly reduced aMMP-8 levels. This was also clearly and significantly corroborated by the improvement in their periodontal parameters. Our findings are in concordance with and further extend the recent study by Heikkinen et al. [52] among a Finnish population with diabetes, in which some of the periodontal clinical indices significantly improved along with the aMMP-8 test results. Despite a large BOP (%) reduction, the change after SRP was ~50%. We relate this to the pre-SRP BOP (%) of ~30%. It is therefore unlikely that the pre- and post-SRP difference would be so much. Furthermore, residual BOP after an NSPT may be associated with subgingival deposits [65]. Although the CAL was reduced in the diabetes group alone, with a small non-significant reduction in the prediabetes group after NSPT, it should be noted that the latest 2017 periodontitis classification also includes CAL non-periodontitis/non-inflammatory causes like traumatic gingival recession [48]. This may explain the non-significant reduction in CAL in all groups except the diabetes group, which had stage IV periodontitis.

The aMMP-8 levels slightly increased after NSPT in normoglycemic patients but not in prediabetes/diabetes cohorts. This may be explained by the insidious nature of periodontitis with burst periods of disease activity: currently stable, currently unstable, or currently in remission [66]. The prediabetes and diabetes groups were from the endocrinology clinic, unlike the normoglycemia patients who came to the dental clinic with symptomatic periodontal disease. The relatively large variances in the pretreatment BOP (%), HbA1c, and aMMP-8 in normoglycemia may partly reflect increased inflammatory activity with developing periodontitis and explain the non-significant NSPT outcomes. It is noteworthy that the significant decrease in aMMP-8 among the prediabetes group after NSPT will further reduce its destructive collagenolytic activity on the periodontal tissues, thus reducing inflammation. Periodontal therapy may thus delay the switch of prediabetes to diabetes if performed properly and on time. This may thus strengthen dentist–physician collaborations in providing optimum care to prediabetes/diabetes patients. Previous research indicates that 25–37% of individuals with prediabetes will develop type 2 diabetes within 3 to 5 years [67], and 46–70% will have it within their lifetime [68,69]. Furthermore, untreated prediabetes increases the risk of diabetes complications, including diabetic retinopathy, peripheral neuropathy, chronic kidney disease, and cardiovascular disease [69]. Thus, aMMP-8 can monitor prediabetes and diabetes patients and provide early intervention to mitigate progression to diabetes and periodontitis [29,30]. It is tempting to speculate that this delayed prediabetes-to-diabetes shift could be enhanced by aMMP-8 inhibitors, including low-dose doxycycline, dual-light photodynamic therapy, and fermented lingonberry mouthwash [30,44,70].

The current study established the link between periodontitis severity, glycemic status, and aMMP-8 following NSPT, as stage III and IV had statistically significant reductions in both aMMP-8 and HbA1c. These findings further cement the link between periodontitis severity and diabetes. When used to monitor periodontal treatment in patients with severe periodontitis (stage III and IV), an aMMP-8 PoC test was the most efficient and precise discriminator, with an optimal cut-off of 20 ng/mL compared to the total MMP-8, tissue inhibitor of MMPs (TIMP)-1, aMMP-8 RFU activity assay, PMN elastase, calprotectin, and interleukin-6 after 6 weeks of NSPT [19,29,30]. While the <20 ng/mL value was not attained for the aMMP-8 PoC after NSPT in our study, it underscores the need for intermittent, even shorter or more frequent periodontal maintenance visits (every four to six weeks) to monitor patients’ compliance with oral hygiene and reinforce these practices. A recent South African study found that increasing the frequency of recall dental visits and monitoring the treatment results over time improved periodontal treatment outcomes [71].

The current study had some limitations, such as the short (3 month) follow-up period of the participants post-SRP. A prolonged monitoring period with frequent intermittent visits may have increased the NSPT effect on aMMP-8 levels in the glycemic groups, especially prediabetes and diabetes with more severe periodontitis. The normoglycemic group had fewer individuals, necessitating some caution when interpreting the results. The strength of the study includes the fact that, to the best of authors’ knowledge, this is the first published study among Nigerians with prediabetes and diabetes to monitor glycemic control (HbA1c levels) after NSPT using an adjunctive on-time non-invasive aMMP-8 PoC diagnostic mouthrinse test. We also used the latest classification system of periodontitis staging to assess periodontitis severity.

## 5. Conclusions

Non-surgical periodontal therapy significantly improves periodontal health as well as aMMP-8 and glycemic levels in prediabetes/diabetes patients and may delay the switch or progression of prediabetes to diabetes and its complications. It is tempting to speculate that in the future, adjunctive aMMP-8 inhibitors, i.e., low-dose doxycycline [44], plaque control’s modification by dual-light photodynamic therapy, and fermented lingonberry mouthrinse [46,70] could further delay the onset of diabetes. Future studies are warranted to see whether anti-aMMP-8 therapeutic interventions can promote this delay. The aMMP-8 PoC mouthrinse test is a useful adjunctive on-time tool to monitor the periodontal treatment response in prediabetes and diabetes, applicable in dental and especially medical settings. These study outcomes could further strengthen the dentist–physician collaboration between dentistry and medicine, as well as a cooperation between dentistry and the medical laboratory, in the provision of holistic care to patients living with prediabetes and diabetes, through early intervention.

## Figures and Tables

**Figure 1 biomedicines-13-00969-f001:**
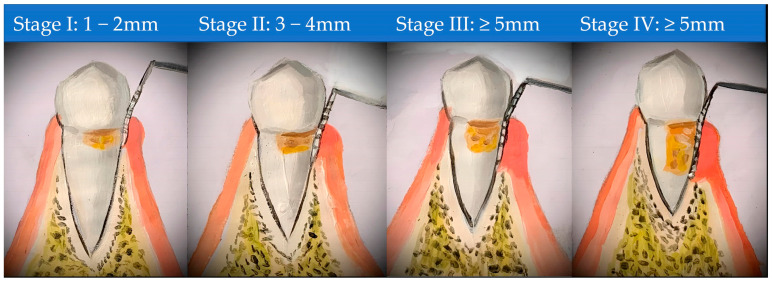
The process of classifying the severity of a patient’s disease. The primary determinant = clinical attachment loss (CAL) at the point of greatest loss (the worst tooth).

**Figure 2 biomedicines-13-00969-f002:**
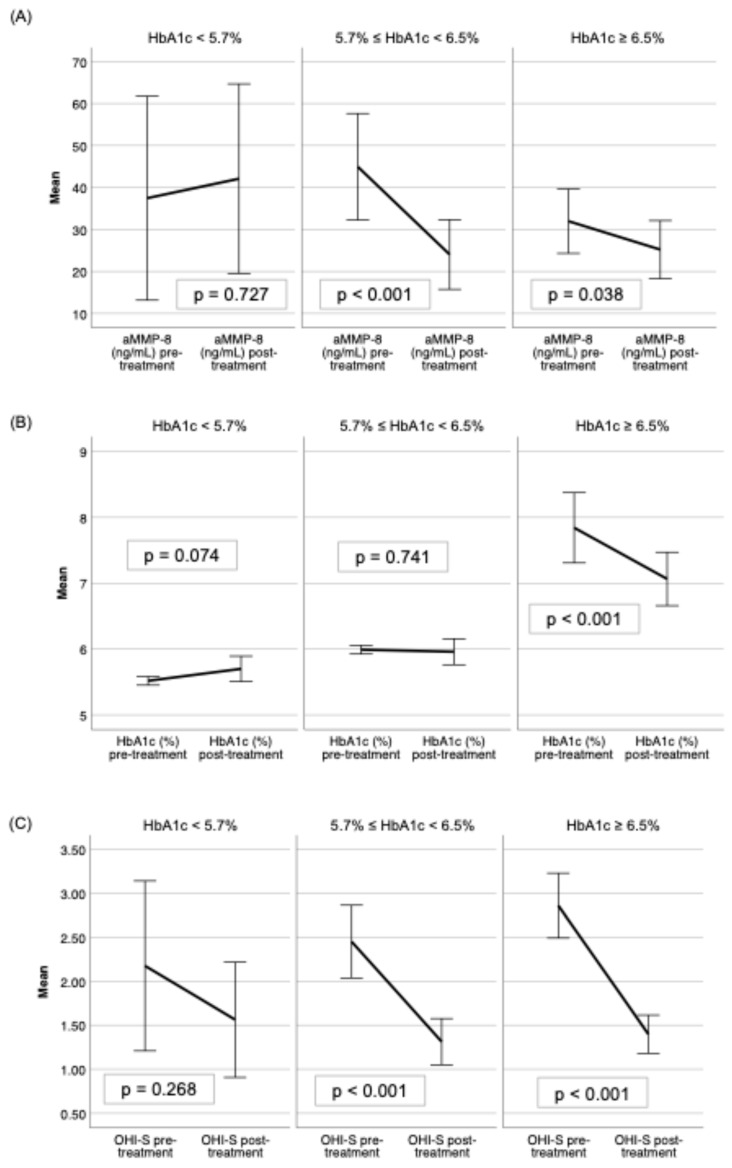
Pre- and post-SRP treatment effect on mean levels with 95% confidence intervals shown of (**A**) aMMP-8 (ng/mL), (**B**) HbA1c (%), and (**C**) OHI-S among normoglycemic (HbA1c < 5.7%), prediabetes (5.7% ≤ HbA1c < 6.5%) and diabetes (HbA1c ≥ 6.5%) groups.

**Figure 3 biomedicines-13-00969-f003:**
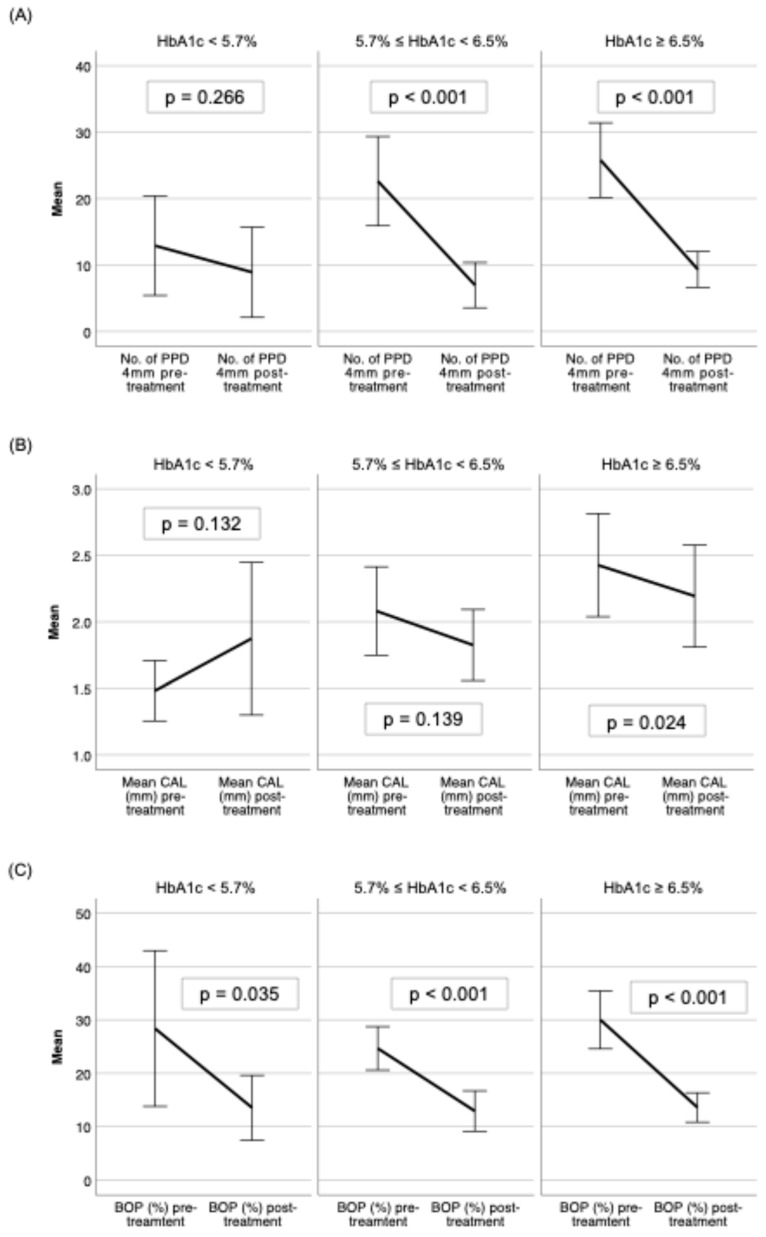
Pre- and post-SRP treatment effect on mean levels with 95% confidence intervals shown of (**A**) PPD ≥ 4 mm, (**B**) mean CAL, and (**C**) BOP (%) among normoglycemic (HbA1c < 5.7%), prediabetic (5.7% ≤ HbA1c < 6.5%), and diabetic (HbA1c ≥ 6.5%) groups.

**Table 1 biomedicines-13-00969-t001:** Sociodemographic characteristics according to glycemic status.

	Normoglycemic	Prediabetes	Diabetes	*p*-Value
	N (%)	N (%)	N (%)	(Fisher’s Exact Test)
Mean age (SD)	50.5 (13.7)	54.7 (13.2)	60.9 (12.3)	0.017 *
Gender				
Male	6 (54.5)	19 (59.4)	21 (46.7)	0.539
Female	5 (45.5)	13 (40.6)	24 (53.3)	
Highest level of education				
Primary	1 (9.1)	1 (3.1)	2 (4.4)	0.444
Secondary	1 (9.1)	12 (37.5)	12 (26.7)	
Tertiary	9 (81.8)	19 (59.4)	31 (68.9)	
Ethnicity				
Yoruba	7 (63.6)	17 (53.1)	24 (53.3)	0.908
Igbo	3 (27.3)	8 (25)	11 (24.4)	
Others	1 (9.1)	7 (21.9)	10 (22.2)	
Religion				
Christianity	8 (72.7)	28 (87.5)	36 (80)	0.495
Islam	3 (27.3)	4 (12.5)	9 (20)	

* Statistical significance. SD: Standard Deviation.

**Table 2 biomedicines-13-00969-t002:** Pre- and post-SRP on periodontal clinical parameters in all groups (N = 88).

Variables	Time	Mean (SD)	Confidence Interval (Paired Differences)		*p*-Value
			Lower Level	Upper Level	
PPD ≥ 4 mm	Pre-SRP–Post-SRP	14.6 (16.2)	11.2	18.0	<0.001 *
	Pre-SRP	23.0 (19.2)			
	Post-SRP	8.4 (9.3)			
CAL (mm)	Pre-SRP–Post-SRP	0.2 (0.8)	−0.01	0.3	0.064
	Pre-SRP	2.2 (1.1)			
	Post-SRP	2.0 (1.1)			
OHI-S	Pre-SRP–Post-SRP	1.2 (1.2)	1.0	1.5	<0.001 *
	Pre-SRP	2.6 (1.2)			
	Post-SRP	1.4 (0.8)			
BOP (%)	Pre-SRP–Post-SRP	14.5 (16.4)	1.7	11.0	<0.001 *
	Pre-SRP	27.9 (16.4)			
	Post-SRP	13.3 (9.6)			

* Statistical significance. SD: Standard Deviation. *p*-values calculated by paired-samples *t*-test (2-sided test).

**Table 3 biomedicines-13-00969-t003:** Pre- and post- SRP aMMP-8 (ng/mL) and HbA1c (%) by periodontitis staging status.

Variables	Time	Mean (SD)	Confidence Interval (Paired Differences)		*p*-Value
			Lower Level	Upper Level	
aMMP-8 (ng/mL)	All groups (N = 88)		4.6	16.4	<0.001 *
	Pre-SRP–Post-SRP	10.5 (27.7)			
	Pre-SRP	37.4 (30.9)			
	Post-SRP	26.9 (24.9)			
	Stage II periodontitis (N = 32)		−0.2	14.5	0.055
	Pre-SRP–Post-SRP	7.2 (20.3)			
	Pre-SRP	29.3 (21.9)			
	Post-SRP	22.2 (22.1)			
	Stage III + IV periodontitis (N = 56)		4.0	20.8	0.004 *
	Pre-SRP–Post-SRP	12.4 (31.2)			
	Pre-SRP	42.0 (34.4)			
	Post-SRP	29.6 (26.2)			
HbA1c (%)	All groups (N = 88)		0.2	0.6	0.001 *
	Pre-SRP–Post-SRP	0.4 (1.0)			
	Pre-SRP	6.9 (1.6)			
	Post-SRP	6.5 (1.2)			
	Stage II periodontitis (N = 32)		−0.08	0.7	0.107
	Pre-SRP–Post-SRP	0.3 (1.1)			
	Pre-SRP	6.8 (1.8)			
	Post-SRP	6.4 (1.2)			
	Stage III + IV periodontitis (N = 56)		0.2	0.7	0.002 *
	Pre-SRP–Post-SRP	0.4 (1.0)			
	Pre-SRP	6.9 (1.5)			
	Post-SRP	6.5 (1.2)			

* Statistical significance. SD: Standard Deviation. *p*-values calculated by paired-samples *t*-test (2-sided test).

## Data Availability

The data that support the findings of this study are available by reasonable request from the corresponding author. The data are not publicly available due to privacy and ethical restrictions.

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
