# Peer review of "Effects of Non-Surgical Periodontal Therapy on Glycemic Control in Prediabetes and Diabetes Patients with Stage II–IV Periodontitis as Monitored by Active-Matrix Metalloproteinase-8 Levels"

_biomedicines, 2025, doi:10.3390/biomedicines13040969_

Round 1
Reviewer 1 Report
Comments and Suggestions for Authors
This manuscript has an important application, whereas it concluded that active matrix metalloproteinase-8 (aMMP-8) levels using point-of-care kits is a useful tool for monitoring periodontal 45 treatment response in prediabetes and diabetes, which can be applied in dental and medical settings. Please respond to the below comments:
1) Are the measured parameters easily used as a routine assay?
2) There are two abbreviations, PoC and POC. Please unify.
3) Please give a small introduction or clarification to PoC test.
4) The introduction has to be shortened.
5) Active matrix metalloproteinase-8 (aMMP-8) should be written only once in both abstract and text.
6) Active matrix metalloproteinase-8 (aMMP-8) concentration is not acceptable. Please use aMMP-8 level(s) in the whole manuscript. In other words, delete concentration and replace it by level.
7) What is the importance of sociodemographic characteristics in Table 1?
8) In the final conclusion, it is better to state that........... a cooperation between dentistry and medical laboratory rather than a cooperation between dentistry and medicine.
Author Response
Comment: 1) Are the measured parameters easily used as a routine assay?
Response: 1) Thank you for your question. Yes, HbA1c is being used as both a laboratory assay and currently a PoC/Chairside routine assay as well. aMMP-8 PoC test is readily and commercially available to dental professionals, including dental practitioners and oral hygienists, in Europe, UK, USA (FDA granted). However, it is not yet a routine assay in many centres. But ongoing research like the current one and several others, that have validated its usefulness, will make it more readily accessible to dental and medical settings in both developed and developing countries like Nigeria.
Comment: 2) There are two abbreviations, PoC and POC. Please unify.
Response: 2) We have unified the two abbreviations, PoC and POC and changed it in the abbreviation list. We have adopted PoC.
Comment: 3) Please give a small introduction or clarification to PoC test.
Response: 3) The aMMP-8 PoC is a safe, rapid, non-invasive diagnostic test that detects the biomarker, aMMP-8 levels for periodontal diseases, using a simple mouthrinse sample. It has been validated and standardized by numerous researches and is commercially available as a lateral flow immunotest stick that is visually (+, -) and quantitatively read by the ORALyzer® Digital Reader System (Dentognostics GmbH, Jena, Germany), with a cut off of 20ng/mL, and a 30 second pre-rinsing. Lines 170-175
Comment: 4) The introduction has to be shortened.
Response: 4) The introduction has been shortened. Thank you
Comment: 5) Active matrix metalloproteinase-8 (aMMP-8) should be written only once in both abstract and text.
Response: 5) Active matrix metalloproteinase-8 has been written only once in both abstract and text.
Comment: 6) Active matrix metalloproteinase-8 (aMMP-8) concentration is not acceptable. Please use aMMP-8 level(s) in the whole manuscript. In other words, delete concentration and replace it by level.
Response: 6) aMMP-8 level(s) has replaced aMMP-8 concentration in the whole manuscript.
Comment: 7) What is the importance of sociodemographic characteristics in Table 1?
Response: 7) The socio-demographic characteristics in Table 1 were included to give an overview of the multi-cultural background of the study participants and to assess the distribution of the study findings. Lines 124-127
Comment: 8) In the final conclusion, it is better to state that........... a cooperation between dentistry and medical laboratory rather than a cooperation between dentistry and medicine.
Response: 8) Thank you for your suggestion. We have incorporated the statement. Lines 385-386
Reviewer 2 Report
Comments and Suggestions for Authors Dear authors, thank you for the opportunity to revise this manuscript. I recommend the following revisions. Abstract- Please, in the Materials and Methods section, either include the phrase “normoglycemia...(45)” in parentheses for clarity or remove it altogether, as participant details are not necessary in the Abstract. If you keep the phrase, it could be rephrased as: “Eighty-eight adults (11 normoglycemic, 32 prediabetic, 45 with type 2 diabetes).”
- “Periodontitis and diabetes are chronic, highly prevalent, interlinked, non-communicable inflammatory diseases that pose substantial public health challenges worldwide” Please change -that pose substantial- with -that present significant- for better readability.
- The sentence “However, synthetic MMP-8 inhibitors like doxycycline can mitigate this effect” could be clearer if rephrased, for example: “However, synthetic inhibitors of MMP-8, such as doxycycline, can help mitigate this effect.”
- “Severe periodontitis affects 23.6% of adults globally [6], while 49% of adults have periodontitis in Nigeria [7]. A recent study reported a higher prevalence of severe periodontitis among PLD (49.1%) compared to non-diabetes (31.8%) [8]”: Please simplify the sentence, for example as follows: “Severe periodontitis affects 23.6% of adults globally, with a notably higher prevalence among patients with diabetes (PLD) at 49.1%, compared to 31.8% in non-diabetic individuals, and 49% in adults in Nigeria” This way, the repetitions of prevalence are eliminated and there is a more direct transition to the next sentence.
- “Proteolytic enzymes, most notably matrix metalloproteinases (MMPs), are the key proteases involved in periodontitis tissue degradation [12]” Please make this section on MMPs more fluid and connected to the previous topic.
- Please, in lines 112-114, when mentioning the additional therapies regarding NSPT, also include the topical agents that should be used prior to professional oral hygiene as plaque disaggregants, such as, for example, the work of Pardo, A. et al. Topical Agents in Biofilm Disaggregation: A Systematic Review and Meta-Analysis. J. Clin. Med. 2024, 13, 2179.
- Please do not include the results in the materials and methods section between lines 130-132, but limit yourselves to presenting the methods used to carry out the research.
- 3.1: “There were 46 males, and 42 females)… ”There is a typing error regarding a parenthesis
- The discussion is well organized, but please include a section on the home evaluation following NSPT in the treatment of diabetic patients, as in the work of Butera, A. et al. Domiciliary Management of Periodontal Indexes and Glycosylated Hemoglobin (HbA1c) in Type 1 Diabetic Patients with Paraprobiotic-Based Toothpaste and Mousse: Randomized Clinical Trial. Appl. Sci. 2022, 12, 8610.
- Please also include the potential clinical implications related to your research.
- Please separate the conclusions from the discussion into a separate paragraph titled "Conclusions."
Author Response
Comment: 1) Abstract: Please, in the Materials and Methods section, either include the phrase “normoglycemia...(45)” in parentheses for clarity or remove it altogether, as participant details are not necessary in the Abstract. If you keep the phrase, it could be rephrased as: “Eighty-eight adults (11 normoglycemic, 32 prediabetic, 45 with type 2 diabetes).”
Response: 1) The authors have rephrased the statement from “Eighty-eight adults; normoglycemia (11), prediabetes (32), type 2 diabetes (45)” to “Eighty-eight adults (11 normoglycemic, 32 prediabetic, 45 with type 2 diabetes).” Lines 33-34
Comment: 2) Introduction “Periodontitis and diabetes are chronic, highly prevalent, interlinked, non-communicable inflammatory diseases that pose substantial public health challenges worldwide” Please change -that pose substantial- with -that present significant- for better readability.
Response: 2) We have replaced “-that pose substantial-“ to “-that present significant-“ Line 52
Comment: 3) The sentence “However, synthetic MMP-8 inhibitors like doxycycline can mitigate this effect” could be clearer if rephrased, for example: “However, synthetic inhibitors of MMP-8, such as doxycycline, can help mitigate this effect.”
Response: 3) We have rephrased the sentence from “However, synthetic MMP-8 inhibitors like doxycycline can mitigate this effect” to “However, synthetic inhibitors of MMP-8, such as doxycycline, can help mitigate this effect.” Lines 95-96.
Comment: 4) “Severe periodontitis affects 23.6% of adults globally [6], while 49% of adults have periodontitis in Nigeria [7]. A recent study reported a higher prevalence of severe periodontitis among PLD (49.1%) compared to non-diabetes (31.8%) [8]”: Please simplify the sentence, for example as follows: “Severe periodontitis affects 23.6% of adults globally, with a notably higher prevalence among patients with diabetes (PLD) at 49.1%, compared to 31.8% in non-diabetic individuals, and 49% in adults in Nigeria” This way, the repetitions of prevalence are eliminated and there is a more direct transition to the next sentence.
Response: 4) We have rephrased the statement from “Severe periodontitis affects 23.6% of adults globally [6], while 49% of adults have periodontitis in Nigeria [7]. A recent study reported a higher prevalence of severe periodontitis among PLD (49.1%) compared to non-diabetes (31.8%) [8]” to “Severe periodontitis affects 23.6% of adults globally [3], with a notably higher prevalence among diabetics at 49.1%, compared to 31.8% in non-diabetics in Nigeria [4].” Lines 59-61.
Comment: 5) “Proteolytic enzymes, most notably matrix metalloproteinases (MMPs), are the key proteases involved in periodontitis tissue degradation [12]” Please make this section on MMPs more fluid and connected to the previous topic.
Response: 5) The paragraph on MMPs has been linked to the preceding section to make it more fluent. “In addition to cytokines, proteolytic enzymes, particularly matrix metalloproteinases (MMPs), are critical proteases that contribute to periodontal tissue degradation [5, 6]”. Lines 72-74
Comment: 6) Please, in lines 112-114, when mentioning the additional therapies regarding NSPT, also include the topical agents that should be used prior to professional oral hygiene as plaque disaggregants, such as, for example, the work of Pardo, A. et al. Topical Agents in Biofilm Disaggregation: A Systematic Review and Meta-Analysis. J. Clin. Med. 2024, 13, 2179.
Response: 6) The authors are very grateful for this insightful contribution and have included the citation. Furthermore, Pardo et al [7], recently reported the application of topical biofilm dissagregating agents during NSPT to improve clinical and microbiological outcomes. Lines 98-100
Comment: 7) M&M: Please do not include the results in the materials and methods section between lines 130-132, but limit yourselves to presenting the methods used to carry out the research.
Response: 7) The repeated results have been removed from line 130-132 in the materials and methods section as suggested by the reviewer. Lines 124-125
Comment: 8) Results 3.1: “There were 46 males, and 42 females)… ”There is a typing error regarding a parenthesis
Response: 8) The parenthesis “....and 42 females) has been removed and the sentence now reads “There were 46 males, and 42 females, aged.....” Line 225
Comment: 9) Discussion The discussion is well organized, but please include a section on the home evaluation following NSPT in the treatment of diabetic patients, as in the work of Butera, A. et al. Domiciliary Management of Periodontal Indexes and Glycosylated Hemoglobin (HbA1c) in Type 1 Diabetic Patients with Paraprobiotic-Based Toothpaste and Mousse: Randomized Clinical Trial. Appl. Sci. 2022, 12, 8610.
Response: 9) A statement on the beneficial effect of some of home remedies following NSPT has been added while incorporating the suggested work of Butera et al. Thank you for this enriching recommendation. Lines 287-290.
Comment: 10) Please also include the potential clinical implications related to your research.
Response: 10) The clinical implications related to our research have been included. Lines 309-310
Comment: 11) Conclusions Please separate the conclusions from the discussion into a separate paragraph titled "Conclusions."
Response: 11) The “Conclusions” have been separated from the discussion and placed in a different paragraph. Lines 385-386.
Reviewer 3 Report
Comments and Suggestions for Authors
congratulations on the paper
I suggest
On lines 197 and 198 looks like there's an extra space before 'using chlorhexidine mouthrinse for irrigation.' To keep the formatting clean and make it easier to read that space should be removed.
Author Response
Comment: congratulations on the paper
I suggest
On lines 197 and 198 looks like there's an extra space before 'using chlorhexidine mouthrinse for irrigation.' To keep the formatting clean and make it easier to read that space should be removed.
Response: The authors thank the reviewer for the kind comments. We have removed the extra space before “using chlorhexidine mouthrinse for irrigation.” To make it easier to read and more fluid. Lines 200-202.
Reviewer 4 Report
Comments and Suggestions for Authors
Dear authors,
This manuscript discusses the effects of non-surgical periodontal therapy on glycemic control and active-matrix metalloproteinase-8 levels in patients with prediabetes and diabetes who have stage II-IV periodontitis.
Your work on the effects of non-surgical periodontal therapy on glycemic control and active-matrix metalloproteinase-8 levels in patients with prediabetes and diabetes who have stage II-IV periodontitis is not only interesting but also of great value. With periodontal disease affecting almost 50% of the global population and the increasing prevalence of diabetes mellitus, your research is crucial in addressing these global health problems. However, some concerns with the report need to be addressed.
Abstract
The conclusion written in the abstract does not reflect the aim of the study; it is an additional observation. Please try to replace it.
Introduction
The introduction is well structured and provides the reader with sufficient background information to understand the study.
However, one observation I will make and is valid for the entire manuscript—according to the journal guidelines for manuscript preparation, the reference numbers in the text must be written as follows: [1-3], not [1] [2][3]. Please revise this issue.
Materials and Methods
Presents detailed information and provides the possibility of reproducing the study.
However, some problems can be observed in this section:
- Are patients in the diabetes group insulin or non-insulin-dependent?
- What were the indications for maintaining home care for oral hygiene?
- For how long and in what dosage was adjuvant systemic antibiotic therapy administered?
- How was the sample size calculated?
- A statistical power test would have helped confirm the robustness of the results.
- On what bases were parametric/nonparametric tests chosen?
- What are the parameters considered for the statistical analysis?
Discussion
This section provides a comprehensive explanation for the results obtained, refers to previous studies, and compares the results obtained with similar research, strengthening the validity of the observations. However, there is no explanation of how systemic antibiotic therapy would influence the obtained results.
References
Please write references according to the journal's requirements (as recommended by the ACS style guide).This manuscript includes excessive self-citations (13 if I counted them all). Please exclude the self-citations or replace them with studies relevant to the current study.
Author Response
Comment: 1) Abstract: The conclusion written in the abstract does not reflect the aim of the study; it is an additional observation. Please try to replace it.
Response: 1) The conclusion has been revised to reflect the aim of the study. A more reflective conclusion has replaced the old one. Thank you. Lines 44-46.
Comment: 2) Introduction
The introduction is well structured and provides the reader with sufficient background information to understand the study. However, one observation I will make and is valid for the entire manuscript—according to the journal guidelines for manuscript preparation, the reference numbers in the text must be written as follows: [1-3], not [1] [2][3]. Please revise this issue.
Response: 2) Thank you for your kind comments. The references have been revised accordingly. Thank you.
Materials and Methods
Presents detailed information and provides the possibility of reproducing the study. However, some problems can be observed in this section:
Comment: 3) Are patients in the diabetes group insulin or non-insulin-dependent?
Response: 3) The patients are non-insulin dependent. For clarity, a sentence, has been included to reflect this. Line 141.
Comment: 4) What were the indications for maintaining home care for oral hygiene?
Response: 4) The oral hygiene home care was to optimize the outcome of the NSPT and minimize biofilm buildup by ensuring patient compliance. Lines 202-203.
Comment: 5) For how long and in what dosage was adjuvant systemic antibiotic therapy administered?
Response: 5) The adjunct systemic antibiotic was for 5 days and the dosage was Caps Amoxicillin 500mg 8hourly and was combined with Tabs Metronidazole 400mg 8 hourly for synergistic effect. Lines 205-206.
Comment: 6) How was the sample size calculated?
Response: 6) A sample size calculation for a paired t-test to evaluate the treatment effect of NSPT was performed (G*Power 3.1), which revealed that a total of 27 patients were required to reach 80% power with a medium effect size 0.50, and a significance level of 5%. This was added by 10% compensation for a possible missing data, which resulted in a total sample size of 30.
Based on Keskin et al, we estimated/expected medium to large effect size for this study and used medium effect size to not underestimate the sample size. Lines 147-153.
Comment: 7) On what bases were parametric/nonparametric tests chosen?
Response: 7) Based on the central limit theorem, sample sizes of at least 30 can be assumed to be normal so that parametric t-tests for paired/independent samples can be used. Fisher's exact test was used for categorical variables in this study based on that it functions well no matter what sample size is used. Lines 216-220.
Comment: 8) What are the parameters considered for the statistical analysis?
Response: 8) We used aMMP-8, HbA1c (%) and Number of PPD sites, CAL, OHI-S, and BOP (%) for the statistical analysis. Lines 213-215.
Comment: 9) Discussion
This section provides a comprehensive explanation for the results obtained, refers to previous studies, and compares the results obtained with similar research, strengthening the validity of the observations. However, there is no explanation of how systemic antibiotic therapy would influence the obtained results.
Response: 9) We thank you for this important observation. We have included a statement of how the systemic antibiotic therapy could influence the obtained results in our study. Lines 283-285.
Comment: 10) References
Please write references according to the journal's requirements (as recommended by the ACS style guide).This manuscript includes excessive self-citations (13 if I counted them all). Please exclude the self-citations or replace them with studies relevant to the current study.
Response: 10) We have revised the references and removed the excessive and unnecessary self-citations to the bearest minimum that is recommended by the ACS style and that are relevant to the current study. We have also included more references from other authors of relevance to the current study and that support some of our statements. Thank you
Round 2
Reviewer 1 Report
Comments and Suggestions for Authors
No further comments. The authors have a good response.
Comments on the Quality of English LanguagePlease kindly request one round of English editing for authors.
Author Response
Comment 1: The English could be improved to more clearly express the research.
Response 1: Thank you very much for your kind comments. A round of English editing has been done particularly in the introduction section (highlighted in red color). Please see Line numbers: 51-55, 58-64, 84-97, 103-109.
Reviewer 2 Report
Comments and Suggestions for Authors
The manuscript is suitable for publication
Author Response
Comment 1: The manuscript is suitable for publication
Response 1: Thank you very much for your kind comments. The authors thank the reviewer immensely for their very instructive assessment of our manuscript.
Reviewer 4 Report
Comments and Suggestions for Authors
The improvements made to the text help increase the scientific impact, and I appreciate the authors' effort in this regard. I noticed that the number of self-citations was reduced from 21 to 13. Even so, there are too many self-citations, at least 13. I recommend that they be removed and/or replaced with others relevant to the study. (reference 23, 24, 28, 29, 31, 34, 40, 42, 46, 47, 48, 49, 52​).
Author Response
General Comment: This manuscript discusses the effects of non-surgical periodontal therapy on glycemic control and active-matrix metalloproteinase-8 levels in patients with prediabetes and diabetes who have stage II-IV periodontitis.
Your work on the effects of non-surgical periodontal therapy on glycemic control and active-matrix metalloproteinase-8 levels in patients with prediabetes and diabetes who have stage II-IV periodontitis is not only interesting but also of great value. With periodontal disease affecting almost 50% of the global population and the increasing prevalence of diabetes mellitus, your research is crucial in addressing these global health problems. However, some concerns with the report need to be addressed.
General Response: The authors thank the reviewer immensely for their very instructive assessment of our manuscript.
Comment 1: Abstract: The conclusion written in the abstract does not reflect the aim of the study; it is an additional observation. Please try to replace it.
Response 1: The conclusion has been revised to reflect the aim of the study. A more reflective conclusion has replaced the old one. Thank you. Line 44-46.
Comment 2: Introduction: The introduction is well structured and provides the reader with sufficient background information to understand the study.
However, one observation I will make and is valid for the entire manuscript—according to the journal guidelines for manuscript preparation, the reference numbers in the text must be written as follows: [1-3], not [1] [2][3]. Please revise this issue.
Response 2: The authors thank the reviewer for this very important observation and have corrected the reference numbers in the text throughout the entire manuscript. Please see Line numbers 53, 69, 74, 77, 81, 86, 91, 99, 101, 105, 175, 185, 274, 285, 296, 315, 353, 357, 360, 368, 391.
Comment 3: Materials and Methods: Presents detailed information and provides the possibility of reproducing the study. However, some problems can be observed in this section:
Are patients in the diabetes group insulin or non-insulin-dependent?
Response 3: The patients are non-insulin dependent and had type 2 diabetes mellitus. For clarity, a sentence has been included to reflect this. Line 141.
Comment 4: What were the indications for maintaining home care for oral hygiene?
Response 4: The oral hygiene home care was to optimize the outcome of the NSPT and minimize biofilm buildup by ensuring patient compliance. Please see lines 202-203.
Comment 5: For how long and in what dosage was adjuvant systemic antibiotic therapy administered?
Response 5: The adjunct systemic antibiotic was for 5 days, and the dosage was Caps Amoxicillin 500 mg 8 hourly and was combined with Tabs Metronidazole 400 mg 8 hourly for a synergistic effect. Please see lines 205-206.
Comment 6: How was the sample size calculated? A statistical power test would have helped confirm the robustness of the results.
Response 6: A sample size calculation for a paired t-test to evaluate the treatment effect of NSPT was performed (G*Power 3.1), which revealed that a total of 27 patients were required to reach 80% power with a medium effect size 0.50, and a significance level of 5%. This was added by 10% compensation for a possible missing data, which resulted in a total sample size of 30. Please see line 147-153.
Comment 7: On what bases were parametric/nonparametric tests chosen?
Response 7: Based on the central limit theorem, that sample sizes of at least 30 can be assumed to be normal, parametric t-tests for paired/independent samples were used. Fisher's exact test was used for categorical variables in this study since it functions well no matter what sample size is used. Please see lines 216-220.
Comment 8: What are the parameters considered for the statistical analysis?
Response 8: The parameters analysed for the study were aMMP-8, HbA1c (%), (Number of PPD sites, CAL, OHI-S, and BOP (%). Please see lines 213-215.
Comment 9: Discussion: This section provides a comprehensive explanation for the results obtained, refers to previous studies, and compares the results obtained with similar research, strengthening the validity of the observations. However, there is no explanation of how systemic antibiotic therapy would influence the obtained results.
Response 9: We thank you for this important observation. We have included a statement of how systemic antibiotic therapy could have influenced the results obtained in our study. Please see lines 280-285.
Comment 10: References: Please write references according to the journal's requirements (as recommended by the ACS style guide). This manuscript includes excessive self-citations (13 if I counted them all). Please exclude the self-citations or replace them with studies relevant to the current study.
Response 10: We have written references according to the journal’s requirements as recommended by the ACS style guide. We have also removed unnecessary self-citations and cited those that are relevant to the current study.
In addition, we have included relevant references from other authors to support some of our statements. Please see the Reference list. Thank you sir/ma.